# Influence of Cooling Rate on Crystallization Behavior of Semi-Crystalline Polypropylene: Experiments and Mathematical Modeling

**DOI:** 10.3390/polym14173646

**Published:** 2022-09-02

**Authors:** Yeyuan Hu, Yang Liao, Yanyan Zheng, Kosuke Ikeda, Ryoji Okabe, Ruifen Wu, Ryota Ozaki, Jun Xu, Qingyan Xu

**Affiliations:** 1Key Laboratory for Advanced Materials Processing Technology, Ministry of Education, School of Materials Science and Engineering, Tsinghua University, Beijing 100084, China; 2Advanced Materials Laboratory of Ministry of Education, Department of Chemical Engineering, Tsinghua University, Beijing 100084, China; 3Composite Laboratory Research & Innovation Center Mitsubishi Heavy Industries, Ltd., Nagasaki 8510392, Japan

**Keywords:** cooling rate, crystallization, polypropylene, Flash DSC, PVT model

## Abstract

As crystallization behavior has a great effect on the injection molding process, the flash differential scanning calorimetry (FSC) method was employed to study the influence of cooling rate on the crystallization behavior of a semi-crystalline polypropylene (PP). As the experimental results show, crystallization temperatures (onset crystallization temperature and maximum crystallization temperature) and crystallinity decrease as the cooling rate increases. In addition, the corresponding mathematical models were established to describe the relationship between the crystallization temperatures/crystallinity and the cooling rate. A revised Tait equation was also carried out based on the mathematical models.

## 1. Introduction

In the last few years, polymers such as polypropylene (PP) have attracted broad attention in scientific research and industrial applications [1,2,3]. With the widespread application of polymers, not only the macro properties can be improved but also the microscopic behaviors should be explored on a deeper level. Moreover, a better understanding of the microscopic behaviors can also be beneficial to an improvement in properties. One of the microscopic behaviors is crystallization behavior, which is closely related to the cooling rate [4,5].

Crystallization behavior’s dependence on cooling rate is necessary to be explored, which affects the modeling of the correlation between the pressure, specific volume and temperature (PVT) of polymers and then influences shrinkage prediction and industrial production [6,7]. The crystallization behavior of polymers mainly includes crystallization temperatures (onset crystallization temperature and maximum crystallization temperature) and crystallinity, and these exert a tremendous influence on polymers’ mechanical properties. Thus, the modulus of elasticity, yield stress and impact resistance are closely related to crystallization behavior [8,9,10]. For example, the PVT model used in the software MoldFlow is the two-domain Tait equation of state (EoS) [11]. However, the cooling rate’s effect is not considered in EoS, which means that the influence of cooling rate in the simulation of the injection molding process by MoldFlow has not been taken into account. As a result, it is necessary to improve the simulation in MoldFlow and thus enhance the products’ quality.

There are already many studies about the relationship between crystallization behavior and cooling rate [12,13,14,15]. Peters et al. [16] used dilatometry (PVT) to reveal the crystallization kinetics and the resulting morphology of isotactic polypropylene homopolymer as a consequence of the combination of non-isothermal cooling at elevated (isobaric) pressure and the application of shear flow. Suplicz et al. [12] proved that the cooling rate has significant influence on the crystallinity of the compounds with a polypropylene (PP) matrix which covered a small range of cooling rate. Wang et al. [13,14] considered the influences of starting temperature and cooling/heating rates and established a PVT model to determine the specific volume evolution of polymers, whereas the measured temperature and pressure were not the exact ones of melting resin. A PVT model for semi-crystalline polymers considering the cooling rate was also proposed by Zuidema et al. [15]. However, current research mostly investigates the low cooling rate range, which is lower than that of the real injection molding process. In addition, the cooling rate observed in the real injection molding process will depend on the thickness of the specimen, mold and melt temperatures. It should also be considered that in an injection-molded specimen with thickness higher than 100 microns a thermal gradient across the thickness will occur and the cooling rate will be different across the specimen thickness.

As far as we are aware, the crystallizing curves can be obtained by the differential scanning calorimetry (DSC) tests at different cooling rates, and the crystallization temperatures can also be calculated from the DSC curves [17,18,19]. However, there is a limitation of the cooling rate range: the common DSC can only realize low cooling rates which have a difference from the real injection molding process [20]. To explore the high cooling rate’s field, the Flash DSC measurement which can reach high degrees of cooling and heating rates should be adopted [21,22]. There are also some studies showing the crystallization behavior and structure formation of isotactic polypropylene at different cooling rates by using the Flash DSC method [23]. In these studies, it is very well described that at cooling at rates higher than 150 K/s a second exothermic event is detected. This exothermic event is related to the formation of a mesophase which forms at lower temperatures. To further explore the quantitative relationship between cooling rate and crystallization behavior, we would like to adopt some more investigations.

In this paper, the Flash DSC measurement is carried out to explore the relationship between the cooling rate and crystallization behavior of polypropylene (PP). A semi-crystalline PP is tested. A new PVT model considering the wide range of cooling rate is also proposed in this study.

## 2. Materials and Methods

### 2.1. Materials and Equipment

A semi-crystalline polypropylene (Talc filled PP, SABIC, Riyadh, Saudi Arabia) was used in the experiments. The melt flow rate (MFR) of the PP at 230 °C and 2.16 kg was 18 g/10 min, and the density was 1239 kg/m^3^ at room temperature. The material was used as received and no purification was conducted.

The Flash DSC 2+ by Mettler-Toledo, Columbus, OH, USA was employed. This instrument could reach ultra-high cooling rates of up to 40,000 K/s, which achieved the exploration of high cooling rate’s effect on crystallization behavior.

A high-pressure capillary rheometer (Rheograph 25, GÖTTFERT Werkstoff-Prüfmaschinen GmbH, Buchen, Germany) was used to measure the PVT diagrams of the PP. This instrument could realize isothermal or isobaric PVT measurements according to ISO 17744. The long diameter ratio of capillary tube was 25:2 and the sample mass weight of PVT diagram test was 3 kg.

### 2.2. Experimental Procedure

The FSC measurement process is illustrated in Figure 1 and Table 1 shows the detailed cooling and heating rates. FSC was used to measure the non-isothermal crystallization process at 18 cooling rates from 0.1 K/s to 1000 K/s, which will cover most of the cases in injection molding process. Furthermore, the heating rate was 200 K/s to obtain obvious melting peak. To eliminate the thermal history, the sample was firstly heated to 220 °C, then cooled to 0 °C at different cooling rates and finally heated to 220 °C. The crystallization behavior could be explored from the cooling and second heating process for each cooling rate. The cooling scans were used to gain information about temperatures of crystallization, while the heating scans were recorded for analysis of the crystallinity which developed in the prior cooling experiment [21,22,23,24].

Table 2 shows the PVT measurement of the isobaric cooling processes. Isobaric PVT measurements were performed at a low ‘constant pressure’ of 500 bar and the temperature changed stepwise between 250 and 50 °C with cooling rate of 0.05 K/s. Afterwards, the pressure was increased to the second constant level of 1000 bar and the temperature variation was repeated. The third pressure level was 1500 bar and the last pressure level was 1800 bar. The data were recorded in temperature intervals of 2 °C.

## 3. Results

### 3.1. FSC Measurements

The Flash DSC curves of the polymer are shown in Figure 2. It can be observed that the data collected on the cooling process clearly show the formation of two exothermic events, i.e., one at higher and one at lower temperature. Because we are mainly concerned the temperatures at which the crystallization initially appeared, we use the exothermic event at higher temperature to measure the temperatures we need. The onset crystallization temperature (*T_s_*) and maximum crystallization temperature (*T_m_*) at each cooling rate can be obtained from these curves referring to ISO 11357-3:1999 (illustrated in Figure 3): The extrapolated start temperature *T_s_* is where the extrapolated baseline is intersected by the tangent to the curve at the point of inflection and corresponds to the start of the transition, and the peak temperature *T_m_* is the temperature at which the peak reaches a maximum.

Figure 4 shows the *T_s_* and *T_m_* of the polymer. It can be seen that both the onset crystallization temperature and the maximum crystallization temperature of PP decrease rapidly at first as the cooling rate increases and then tend to be stable. The relationship between crystallization temperatures and cooling rate *r* is modeled:(1)$$T_{s}=d_{1}-k_{1}\times r^{t_{1}}$$
(2)$$T_{m}=d_{2}-k_{2}\times r^{t_{2}}$$
where coefficients *d*_1_, *d*_2_, *k*_1_, *k*_2_, *t*_1_ and *t*_2_ can be obtained from the regulation of *T_s_* − *r* and *T_m_* − *r* experimental curves. When the experimental data of *T_s_*/*T_m_* and corresponding cooling rates (*r*) from Flash DSC were acquired, we could draw the scatter diagrams shown in Figure 4. Then the regulation curves (in red) and equations (Equations (1) and (2)) could be obtained by Origin software.

### 3.2. The Effect of Cooling Rate on Crystallinity

As for the calculation of crystallinity, an extrapolation method was used in this study. Crystallinity can be calculated from the DSC measurement of the melting enthalpy conventionally [17,18,19]; however, as the sample in the Flash DSC was too small (~ng) to obtain the accurate mass, crystallinity could not be obtained from FSC in this traditional way. Considering that the crystallinity was proportional to the heat absorbed in the melting process [25], relative crystallinity could be calculated by using the melting heat at each cooling rate to divide the melting heat at a very low cooling rate. Subsequently, the melting heat at very low cooling rate (approaching zero) was able to be extrapolated from the heat capacity curve of the area of melting peak Q with the cooling rate *r*. As a result, an extrapolation method was used to calculate the relative crystallinity.

Figure 5 shows the crystallinity of the polymer calculated by the extrapolation method. It shows that the crystallinity of PP decreases as the cooling rate increases. It decreases dramatically if the rate is lower than 200 K/s; after that, the crystallization changes little with increasing cooling rate. The relationship between the crystallinity *X_c_* and cooling rate *r* is modeled:(3)$$X_{c}=e+f\times g^{r}$$
where coefficients *e*, *f* and *g* can be obtained from the regulation of the *X_c_* − *r* curve.

### 3.3. PVT Diagram with No Consideration of Cooling Rate

The PVT diagram at a fixed cooling rate of 0.05 K/s of the isobaric cooling processes is presented in Figure 6. The data points were measured by a high-pressure capillary rheometer and the continuous lines were calculated according to the two-domain Tait equation [26]. Table 3 shows the corresponding parameters of the two-domain equation of PP tested.

## 4. Discussion

### 4.1. Crystallization Behavior

Figure 2 shows the thermal behavior observed on both cooling and heating. The data collected on cooling clearly show the formation of two exothermic events, i.e., one at higher and one at lower temperature. As the reference shows [23], we inferred that the exothermic event at lower temperature was related to the formation of a mesophase when the cooling rate was high. It is illustrated in Figure 4 that the decrease rate of crystallization temperatures (*T_s_* and *T_m_*) is reduced when the cooling rate increases. Moreover, the tendency of crystallinity with cooling rate is similar to that of the crystallization temperatures with cooling rate. This is because with the cooling rate increasing, the time for PP to crystallize is shortened. Then there is not enough time to form crystals, and thus the crystallinity will decrease accordingly [27,28]. As a result, the increase in cooling rate will decrease the crystallization temperatures, which means that it is harder for crystallization.

### 4.2. Tait Equation Considering Cooling Rate

A common model used in simulating the injection molding process is the two-domain Tait equation of state (EoS), in which a discontinuity is at the transition temperature between the models for the molten and solid states (two domains). It is also used in commercial software such as MoldFlow. The equation is shown as below [26]:(4)$$v\left(p,T\right)=v_{0}\left(T\right)\times \left\{1-C\times \ln\left[1+\frac{p}{B\left(T\right)}\right]\right\}+v_{t}\left(p,T\right)$$
(5)$$T_{t}\left(p\right)=b_{5}+b_{6}\times p$$
(6)$$\bar{T}=T-b_{5}$$
when *T* > *T_t_*(*p*),
(7)$$v_{0}\left(0,T\right)=b_{1m}+b_{2m}\times \bar{T}$$
(8)$$B\left(T\right)=b_{3m}\times \exp\left(-b_{4m}\times \bar{T}\right)$$
(9)$$v_{t}\left(p,T\right)=0$$
when *T* < *T_t_*(*p*),
(10)$$v_{0}\left(0,T\right)=b_{1s}+b_{2s}\times \bar{T}$$
(11)$$B\left(T\right)=b_{3s}\times \exp\left(-b_{4s}\times \bar{T}\right)$$
(12)$$v_{t}\left(p,T\right)=b_{7}\times \exp\left(b_{8}\cdot \bar{T}-b_{9}\times p\right)$$
in which *C* is a universal constant of 0.0894 and 13 materials constants are included in Equations (4)–(12): *b*_1*m*_, *b*_1*s*_, *b*_2*m*_, *b*_2*s*_, *b*_3*m*_, *b*_3*s*_, *b*_4*m*_, *b*_4*s*_, *b*_5_, *b*_6_, *b*_7_, *b*_8_ and *b*_9_. *V*_0_ is the specific volume when the pressure is zero, *b*_1_ (*b*_1*m*_, *b*_1*s*_) and *b*_2_ (*b*_2*m*_, *b*_2*s*_) are the coefficients to represent the dependence of *v*_0_ on pressure and temperature, *b*_3_ (*b*_3*m*_, *b*_3*s*_) and *b*_4_ (*b*_4*m*_, *b*_4*s*_) are the materials constants, *v_t_* is the specific volume damage caused by crystallization, *T_t_* is the transition temperature, and *b*_5_ represents the glass transition temperature under zero pressure. Parameter *b*_6_ is the material constant for the dependence of the glass transition temperature on pressure, and *b*_7_, *b*_8_ and *b*_9_ are the particular parameters of semi-crystalline polymers that describe the form of the state transition [29].

The corresponding parameters of the two-domain equation of PP tested are presented in Table 3. This model describes the relationship between the pressure *P*, specific volume *V* and the temperature. It can be used for predicting the resin flow behavior in the filling phase and shrinkage deformation behavior in the cooling phase. However, there is no consideration of the cooling rate, while in the case of a crystalline thermoplastic resin, the crystallization temperature varies depending on the cooling rate [29].

As the previous studies show, the related parameters on the Tait EoS with cooling rate effects are *b*_5_, *b*_1*s*_ and *b*_1*m*_ [29]. Parameter *b*_5_ represents the crystallization temperature (*T_s_*) for semi-crystalline polymers and the glass transition temperature (*T_g_*) for amorphous polymers under zero pressure. *b*_1*m*_ and *b*_1*s*_ should be the intercept of the melt-state *V*–*T* line and the solid-state *V*–*T* line when the pressure is zero, respectively [26].

In our experimental results, we can obtain the relationship between the onset crystallization temperature *T_s_* and cooling rate *r* which can be used for modifying the parameter *b*_5_:(13)$$b_{5}=T_{s}=k_{1}-d_{1}\times r^{t_{1}}$$

Because the crystallinity calculated in our study was crystallinity in the whole crystallization process of each sample, this meant that the polymer was totally in solid state. As a result, we considered modifying *b*_1*s*_ (which was in sold state) according to the crystallinity.

It is recorded that *X_c_* can be calculated through the equation shown below [30]:(14)$$X_{c}=\left(V_{a}-V\right)/\left(V_{a}-V_{c}\right)$$
where *V_a_* represents the specific volume of polymer in a completely amorphous state and *V_c_* means the specific volume of polymer in a perfectly crystalline state. Additionally, for most polymers *V_a_*/*V_c_* ≈ 1.13 [30], we can obtain the specific volume *V* though the equation below:(15)$$V=V_{a}/\left(1+0.13\times X_{c}\right)$$
where *V_a_* can be obtained from the *V*–*T* curve of our materials as shown in Figure 6, in which *V_a_* is the specific volume when the polymer is completely melting (*T_c_*) at 0 bar. Therefore, the specific volume *V*_0_ at *P* = 0 and *T* = 0 °C can be described as Equation (14) shows and it is also equal to *b*_1*s*_, which is the intercept of the solid-state *V*–*T* line when the pressure is zero:(16)$$b_{1s}=V_{0}=V_{a}/\left(1+0.13\times X_{c}\right)$$

As a result, we made modifications on *b*_5_ and *b*_1*s*_ through considering the influence of cooling rate.

## 5. Conclusions

In this study, the relationship between the cooling rate and crystallization behavior of polypropylene has been studied by a method of Flash DSC measurement.

It can be concluded that, with the increase in cooling rate, both the crystallization temperatures and the crystallinity decrease rapidly in earlier stages and then change a little. This shows that the cooling rate does have an obvious effect on the crystallization behavior of PP.

As the Tait equation of state was modified by taking a wide range of cooling rates into consideration, we could combine the simulation of the injection molding process in MoldFlow software in the next step. Some injection molding experiments could also be conducted to validate our modified model.

## Figures and Tables

**Figure 1 polymers-14-03646-f001:**
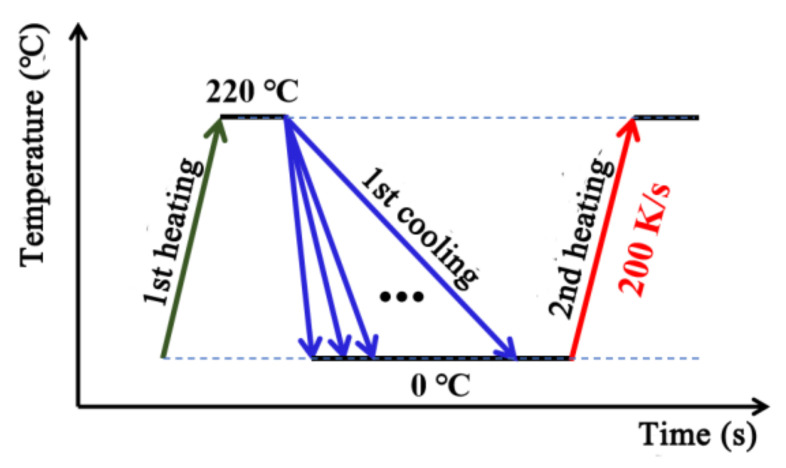
Temperature–time profile of Flash DSC experiments.

**Figure 2 polymers-14-03646-f002:**
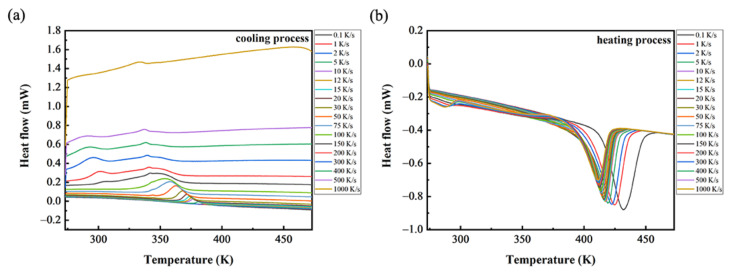
Flash DSC measurements of PP at cooling rates from 0.1 to 1000 K/s: (**a**) cooling process; (**b**) heating process.

**Figure 3 polymers-14-03646-f003:**
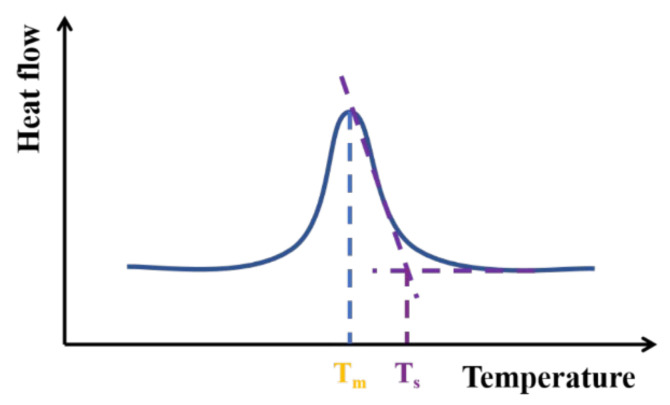
Geometrical relationship between the onset crystallization temperature (*T_s_*) and the maximum crystallization temperature (*T_m_*).

**Figure 4 polymers-14-03646-f004:**
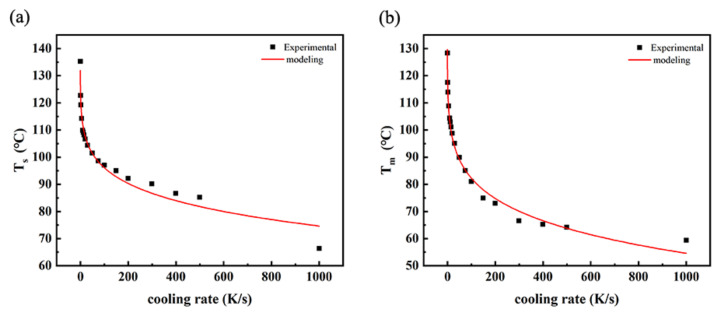
The relationship between the crystallization temperatures ((**a**) *T_s_* and (**b**) *T_m_*) and cooling rate *r* of PP.

**Figure 5 polymers-14-03646-f005:**
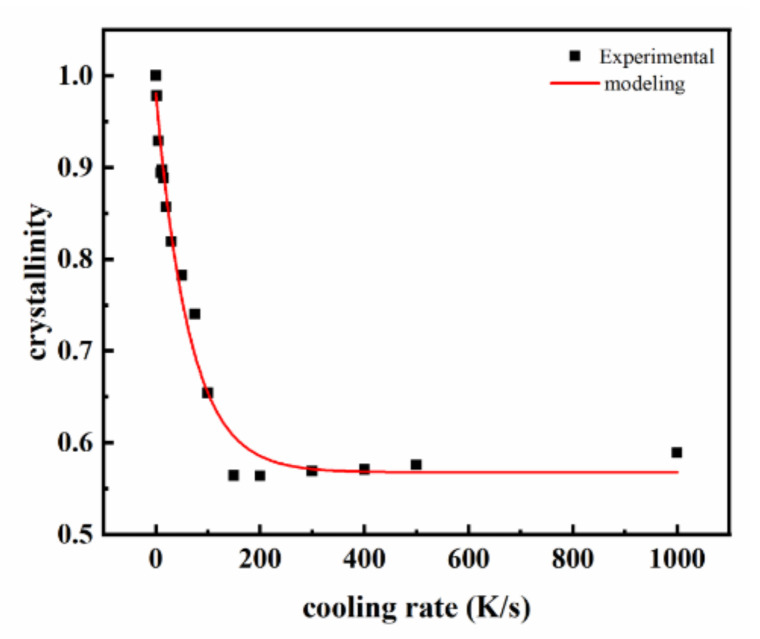
The relationship between crystallinity and cooling rate of PP.

**Figure 6 polymers-14-03646-f006:**
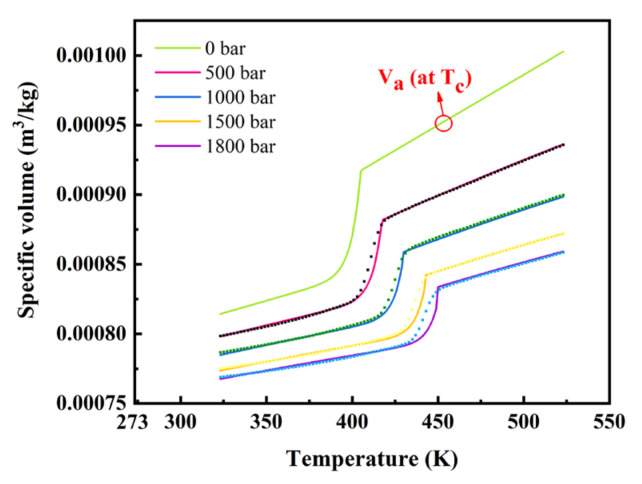
The PVT diagram measured by high-pressure capillary rheometer (in points) and calculated by 2-domain Tait equation (in lines).

**Table 1 polymers-14-03646-t001:** Detailed cooling rates in the FSC measurement process.

Group	1	2	3	4	5	6	7	8	9	10	11	12	13	14	15	16	17	18
1st cooling rate (K/s)	0.1	1	2	5	10	12	15	20	30	50	75	100	150	200	300	400	500	1000

**Table 2 polymers-14-03646-t002:** PVT measurement processes of isobaric cooling.

Temperature (°C)	Pressure (Bar)	Cooling Rate (K/s)
250 → 50	500	0.05
	1000	
	1500	
	1800	

**Table 3 polymers-14-03646-t003:** Coefficients in 2-domain Tait equation for PP.

Coefficient	-
b_5_ (K)	405.2
b_6_ (K/Pa)	2.459 × 10^−7^
b_1m_ (m^3^/Kg)	9.175 × 10^−4^
b_2m_ (m^3^/Kg·K)	7.240 × 10^−7^
b_3m_ (Pa)	7.393 × 10^7^
b_4m_ (1/K)	4.136 × 10^−3^
b_1s_ (m^3^/Kg)	8.434 × 10^−4^
b_2s_ (m^3^/Kg·K)	3.545 × 10^−7^
b_3s_ (Pa)	1.561 × 10^8^
b_4s_ (1/K)	3.042 × 10^−3^
b_7_ (m^3^/Kg)	7.542 × 10^−5^
b_8_ (1/K)	1.834 × 10^−1^
b_9_ (1/Pa)	5.106 × 10^−8^

## Data Availability

Not applicable.

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
