# Peer review of "Influence of Cooling Rate on Crystallization Behavior of Semi-Crystalline Polypropylene: Experiments and Mathematical Modeling"

_polymers, 2022, doi:10.3390/polym14173646_

Round 1

Reviewer 1 Report

Review report

Manuscript ID: polymers-1860790

Manuscript title: Influence of cooling rate on crystallization behavior of semi-crystalline polypropylene: experiments and mathematical modeling

1.     What is the application of the implemented work ? You should explain the objective as it is not clear for example the cooling rates corresponds to which process?

I propose the paper for publication regarding the amount of performed work. However, the objective of the paper should be clearly explained by adding some example regarding the application of the propose modeling..

Best wishes in your work

Author Response

Comments:

“I propose the paper for publication regarding the amount of performed work. However, the objective of the paper should be clearly explained by adding some example regarding the application of the propose modeling.

We appreciate your clear and detailed feedback and hope that the explanation has fully addressed all of your concerns. In the remainder of this letter, we discuss each of your comments individually along with our corresponding responses.

To facilitate this discussion, we first retype your comments in italic font and then present our responses to the comments.

Comment :

What is the application of the implemented work ? You should explain the objective as it is not clear for example the cooling rates corresponds to which process?

Response :

Thank you for your great suggestion! In the previous simulation of injectioin-moulding process using MoldFlow software, the influence of cooling rate on onset crystallization temperature/crystallinity has not been considered in the two-domain Tait equation of state. Our implemented work has established the relationship between onset crystallization temperature/crystallinity and cooling rate. Subsequently the two-domain Tait equation of state was modified with considering the cooling rate’s effect on crystallization. As a result, we can use API Solver in MoldFlow software to replace the original Tait equation with our modified equation, and then our simulation results can be more close to the real injectioin-moulding process.

Reviewer 2 Report

The paper „Influence of cooling rate on crystallization behaviour of semi-crystalline polypropylene: experiments and mathematical modeling“ is covering a very interesting and important topic. The novelty of the contribution is in trying to model the crystallization behaviour of talcum-filled polypropylene blend using data collected from Fast scanning calorimetry and PVT experiments. From this point of view, I would recommend to publish the paper but only after applying major corrections. Here please bellow my recommendations and comments:

1.    The English language and style should be improved. In couple of cases sentences are starting with “And”. I would like to suggest also revision of the following expressions:

-       “was modelling” (line 150, and in another cases): I would propose “was modelled”

-       ..”as the samples in the Flash DSC is too small..” (line 138)

-       “can be gotten” (line 109, 130): such an expression could be replaced by is obtained

-       Crystallization behaviours: I would suggest to the authors to use it in singular form “behaviour”

-       “crystallization start temperature”: I would suggest onset crystallization temperature

-       “Injection process”: the correct term would be injectioin-moulding process

2.    Introduction part:

-       The introduction part needs to be enriched with studies showing the crystallization behaviour and structure formation of isotactic polypropylene at different cooling rates (Quamer Zia, René Androsch, Christoph Schick, etc.). In these studies is very well described that on cooling at rates higher than 150 K/s a second exothermic event is detected. This exothermic event is related to the formation of a mesophase which forms at lower temperature. I believe these studies will help the authors in the discussion of their own experimental data. Beside that the groud of Prof. Gerrit Peters performed PVT analysis on polypropylenes as well. My recommendation would be to mention these studies in the introduction as well and discuss them in the frame of the own work.

3.    Materials

-       In this part, the material of choice is a polypropylene talc filled blend with a density of 1239 kg/m3. Could the authors please mention also the talc concentration of the bland? Perhaps a TGA would help for the determination. I would assume that it is above 25 wt%.

-       Is the talc well dispersed in the sample? Is the talc particle size known?

-       The polypropylene was a homo or random copolymer?

4.    Experimental procedure

-       The experimental part is well described. However, it would be important to provide the sample mass weight used for the experiments done with either FSC or PVT. Specifically for the FSC samples, I would be pleased to know the sample dimensions?

5.    Results

-       Figure 2 is showing according to me two different sets of data. In the top part of the figure, data collected on cooling is presented, while in the bottom part of the figure data collected on heating is shown. My recommendation is to present the data in separate plots and include in the discussion part text about the thermal behaviour observed on both cooling and heating. The data collected on cooling is clearly showing the formation of two exothermic events i.e. one at higher and one at lower temperatures. Could the authors please mention the temperatures at which the events occur? Which exothermic event was considered for the modelling and why the choice was made?.

-       Equations 1 and 2 include plenty of coefficients. Could the authors please explain from which part of the experimental results they are obtained?

-       Could the authors include also the formula used to calculate the relative crystallinity as explained on page 5.

-       Equation 3 contains parameters which meaning is not clear.

6.    Discussion

-       My strong recommendation is to discuss the unique crystallization behaviour of iPP in this part in the light of monoclinic and mesophase formation. It is unclear that at such high talc loading the material still show two exothermic events.

-       Could the authors confirm that the sample selected for the FSC measurements contained talc particles? I would expect that the talc nucleates the iPP and thus supressed the mesophase formation and/or shift the monoclinic phase crystallization at higher temperatures. In Figure 2, the crystallization temperature at the lowest cooling rates is about 105°C, which appears relatively low for a talc filled compound.

-       Eq 4 to 12: Could please the authors give an explanation for each of the factors and constants? What is the physical meaning behind?

-       Line 206 is not clear. “And as the crystallinity at each cooling rate was cooled to 0°C” … please correct.

-       From where it is known that Va/Vc=1.13? (line 213

7.    Conclusions

As mentioned in the beginning, I find novelty in the approach to modify the MoldFlow equation considering data collected from FSC. However, I don’t see that the data supports the conclusions, especially since on cooling two crystallization events are recorded, while in the study only one is concerned.

Author Response

Comments:

“The paper ‘Influence of cooling rate on crystallization behaviour of semi-crystalline polypropylene: experiments and mathematical modeling’ is covering a very interesting and important topic. The novelty of the contribution is in trying to model the crystallization behaviour of talcum-filled polypropylene blend using data collected from Fast scanning calorimetry and PVT experiments. From this point of view, I would recommend to publish the paper but only after applying major corrections.”

We also appreciate your clear and detailed feedback and hope that the explanation has fully addressed all of your concerns. In the remainder of this letter, we discuss each of your comments individually along with our corresponding responses.

To facilitate this discussion, we first retype your comments in italic font and then present our responses to the comments.

Comment 1:

The English language and style should be improved. In couple of cases sentences are starting with “And”. I would like to suggest also revision of the following expressions:

-     “was modelling” (line 150, and in another cases): I would propose “was modelled”

-     ..”as the samples in the Flash DSC is too small..” (line 138)

-     “can be gotten” (line 109, 130): such an expression could be replaced by is obtained

-   Crystallization behaviours: I would suggest to the authors to use it in singular form “behaviour”

-      “crystallization start temperature”: I would suggest onset crystallization temperature

-     “Injection process”: the correct term would be injectioin-moulding process

Response 1:

Thank you for the detailed review. We have carefully and thoroughly proofread the manuscript to correct all the grammar and typos.

Comment 2:

 Introduction part:

-      The introduction part needs to be enriched with studies showing the crystallization behaviour and structure formation of isotactic polypropylene at different cooling rates (Quamer Zia, René Androsch, Christoph Schick, etc.). In these studies is very well described that on cooling at rates higher than 150 K/s a second exothermic event is detected. This exothermic event is related to the formation of a mesophase which forms at lower temperature. I believe these studies will help the authors in the discussion of their own experimental data. Beside that the groud of Prof. Gerrit Peters performed PVT analysis on polypropylenes as well. My recommendation would be to mention these studies in the introduction as well and discuss them in the frame of the own work.

Response 2:

Thanks for your great suggestion on improving the accessibility of our manuscript. We have read literature you recommend carefully. These researches provide me with great reference value and I’d like to mention them in the introduction and discuss them in the frame of the own work. (line 46, 64, 182)

Comment 3:

Materials

-       In this part, the material of choice is a polypropylene talc filled blend with a density of 1239 kg/m3. Could the authors please mention also the talc concentration of the bland? Perhaps a TGA would help for the determination. I would assume that it is above 25 wt%.

-       Is the talc well dispersed in the sample? Is the talc particle size known?

-       The polypropylene was a homo or random copolymer?

Response 3:

Thanks for your great suggestion, however I’m sorry that we have a confidentiality agreement with our cooperative company in which we are not able to public the material’s composition or detailed material brand. The polypropylene was a random copolymer.

Comment 4:

Experimental procedure

The experimental part is well described. However, it would be important to provide the sample mass weight used for the experiments done with either FSC or PVT. Specifically for the FSC samples, I would be pleased to know the sample dimensions?

Response 4:

Thank you for your approval and suggestion. The sample mass weight of PVT diagram test is 3kg. However, the mass of the FSC sample is too small to be weighted and is on the order of ng. This information has been added in the manuscript. (line 154, 87)

Comment 5:

Results

-     Figure 2 is showing according to me two different sets of data. In the top part of the figure, data collected on cooling is presented, while in the bottom part of the figure data collected on heating is shown. My recommendation is to present the data in separate plots and include in the discussion part text about the thermal behaviour observed on both cooling and heating. The data collected on cooling is clearly showing the formation of two exothermic events i.e. one at higher and one at lower temperatures. Could the authors please mention the temperatures at which the events occur? Which exothermic event was considered for the modelling and why the choice was made?.

-      Equations 1 and 2 include plenty of coefficients. Could the authors please explain from which part of the experimental results they are obtained?

-   Could the authors include also the formula used to calculate the relative crystallinity as explained on page 5.

-      Equation 3 contains parameters which meaning is not clear.

Response 5:

Thank you for your comments. We have changed Figure 2 to present the cooling and heating data in separate plots. The discussion part text about the thermal behaviour observed on both cooling and heating has also been added. (line 184) The detailed temperatures description was mentioned. (line 118) The exothermic event at higher temperature was considered for the modelling of Ts and Tm because we concern the temperature when the crystallization behaviour initially appeared and we use the endothermic event on heating process to calculate the crystallinity. (line 151)

The coefficients in Equation 1, 2 and 3 are obtained from the regulation of our experimental curves. (line 142, 167)

Comment 6:

Discussion

-   My strong recommendation is to discuss the unique crystallization behaviour of iPP in this part in the light of monoclinic and mesophase formation. It is unclear that at such high talc loading the material still show two exothermic events.

-   Could the authors confirm that the sample selected for the FSC measurements contained talc particles? I would expect that the talc nucleates the iPP and thus supressed the mesophase formation and/or shift the monoclinic phase crystallization at higher temperatures. In Figure 2, the crystallization temperature at the lowest cooling rates is about 105°C, which appears relatively low for a talc filled compound.

-   Eq 4 to 12: Could please the authors give an explanation for each of the factors and constants? What is the physical meaning behind?

-   Line 206 is not clear. “And as the crystallinity at each cooling rate was cooled to 0°C” … please correct.

-   From where it is known that Va/Vc=1.13? (line 213

Response 6:

Thanks for your recommendation! The text discussing the crystallization behaviour of iPP has been added in the manuscript (line 184). We have also added the physical meaning of each parameter (line 212). The description in line 206 you mentioned has been adjusted (line 236) and we supplied the reference of “Va/Vc=1.13” as well (line 244).

Comment 7:

Conclusions

As mentioned in the beginning, I find novelty in the approach to modify the MoldFlow equation considering data collected from FSC. However, I don’t see that the data supports the conclusions, especially since on cooling two crystallization events are recorded, while in the study only one is concerned.

Response 7:

Thanks for your comment! We have added the text discussing the formation of two exothermic events on cooling process (line 184) and explaining why we choose the crystallization event at higher temperatures on cooling to get Ts/Tm while we choose the heating process to calculate crystallinity in our study (line 118).

We would like to take this opportunity to thank you for all your time involved and this great opportunity for us to improve the manuscript. We hope you will find this revised version satisfactory.

Sincerely,

The Authors

Round 2

Reviewer 1 Report

Good luck

Author Response

Dear reviewer,

Thanks for your patience and supporting!

Reviewer 2 Report

I would like to thank the authors for the prompt correction of the manuscript and taking into consideration the majority of the recommendations. The manuscript is improved and can be considered for publishing, however I still have some minor points for revision, as mentioned bellow:

The words "microscopic behavior" are repeating three times in the first paragraph of the introduction. I would perhaps combine the sentences as following: “With the widespread application of polymers, deeper understanding of the crystallization behaviour in relation to modelling can be beneficial to the improvement of material properties. The crystallization behavior is closely related to the cooling rate as shown elsewhere [4,5].”

Line 34: The verb affect is repeated two times in the same sentence, perhaps once it can be replaced by influence, control, etc.

Line 16 „injection-moulding“ instead of injecting

Line 34 the abbreviation PVT occurs for the first time, thus the complete meaning should be written

Line 40: Can the authors explain shortly what is “the two-domain Tait equation of state”?

Line 56: If the authors make comparison between the studies, it would be worth to mention the cooling rate observed in real injection-moulding process. I guess, it will depend on the thickness of the specimen, mould and melt temperatures. It should also be considered that in an injection-moulded specimen with thickness higher than 100 microns a thermal gradient across the thickness will occur and the cooling rate will be different across the specimen thickness.

Figure 2: I guess there is a typo-mistake in the temperature units; it should be K instead of °C.

Line 131: It is not clear how the coefficients, d2, k1, k2, t1 and t2 can be obtained from the curves on Figure 4? Is the d the intercept and k the slope? Please be more precise.

Line 160: I guess there should be Table 3 instead of Table 2.

Line 171: I would expect that the crystallization temperature drop down fast when the cooling rate is above or faster than 200 K/s. Please correct

Author Response

Comments:

“I would like to thank the authors for the prompt correction of the manuscript and taking into consideration the majority of the recommendations. The manuscript is improved and can be considered for publishing, however I still have some minor points for revision, as mentioned bellow:”

We also appreciate your clear and detailed feedback and hope that the explanation has fully addressed all of your concerns. In the remainder of this letter, we discuss each of your comments individually along with our corresponding responses.

To facilitate this discussion, we first retype your comments in italic font and then present our responses to the comments.

Line 34: The verb affect is repeated two times in the same sentence, perhaps once it can be replaced by influence, control, etc.

Line 16 „injection-moulding“ instead of injecting

Response:

Thank you for the detailed review. We have revised these descriptions (line 35 and line 16).

Line 34 the abbreviation PVT occurs for the first time, thus the complete meaning should be written

Line 40: Can the authors explain shortly what is “the two-domain Tait equation of state”?

Response:

Thank you for your suggestions! We have added the explaining texts respectively (line 34 and line 185).

Line 56: If the authors make comparison between the studies, it would be worth to mention the cooling rate observed in real injection-moulding process. I guess, it will depend on the thickness of the specimen, mould and melt temperatures. It should also be considered that in an injection-moulded specimen with thickness higher than 100 microns a thermal gradient across the thickness will occur and the cooling rate will be different across the specimen thickness.

Response:

Thank you for your suggestions! We have added the discussing text according to your suggestion (line 59).

Figure 2: I guess there is a typo-mistake in the temperature units; it should be K instead of °C.

Response:

Thank you for your reminding. We have revised the typo-mistake in the temperature units in Figure 2.

Line 131: It is not clear how the coefficients, d2, k1, k2, t1 and t2 can be obtained from the curves on Figure 4? Is the d the intercept and k the slope? Please be more precise.

Response:

Thank you for your comment. We have added the more precise text to describe how we got equation 1 and equation 2 (line 140).

Line 160: I guess there should be Table 3 instead of Table 2.

Response:

Thank you for your reminding. We have revised the serial number of Table 3.

Line 171: I would expect that the crystallization temperature drop down fast when the cooling rate is above or faster than 200 K/s. Please correct

Response:

Thank you for your suggestion. We have corrected the expression to avoid ambiguity (line184).

We would like to take this opportunity to thank you for all your time involved and this great opportunity for us to improve the manuscript. We hope you will find this revised version satisfactory.

Sincerely,

The Authors